# Insights into SARS-CoV-2 Persistence and Its Relevance

**DOI:** 10.3390/v13061025

**Published:** 2021-05-29

**Authors:** Belete A. Desimmie, Yonas Y. Raru, Hesham M. Awadh, Peimei He, Samson Teka, Kara S. Willenburg

**Affiliations:** Department of Internal Medicine, Marshall University Joan C. Edwards School of Medicine, Huntington, WV 25701, USA; raru@marshall.edu (Y.Y.R.); awadh@marshall.edu (H.M.A.); hep@marshall.edu (P.H.); teka@marshall.edu (S.T.)

**Keywords:** coronaviruses, SARS-CoV-2, COVID-19, viral persistence, reinfection, long COVID, PASC

## Abstract

Severe acute respiratory syndrome coronavirus 2 (SARS-CoV-2), the causative agent of coronavirus disease 2019 (COVID-19), continues to wreak havoc, threatening the public health services and imposing economic collapse worldwide. Tailoring public health responses to the SARS-CoV-2 pandemic depends on understanding the mechanism of viral replication, disease pathogenesis, accurately identifying acute infections, and mapping the spreading risk of hotspots across the globe. However, effective identification and isolation of persons with asymptomatic and mild SARS-CoV-2 infections remain the major obstacles to efforts in controlling the SARS-CoV-2 spread and hence the pandemic. Understanding the mechanism of persistent viral shedding, reinfection, and the post-acute sequalae of SARS-CoV-2 infection (PASC) is crucial in our efforts to combat the pandemic and provide better care and rehabilitation to survivors. Here, we present a living literature review (January 2020 through 15 March 2021) on SARS-CoV-2 viral persistence, reinfection, and PASC. We also highlight potential areas of research to uncover putative links between viral persistence, intra-host evolution, host immune status, and protective immunity to guide and direct future basic science and clinical research priorities.

## 1. Introduction

In December 2019, SARS-CoV-2, the etiology of COVID-19, first emerged in Wuhan, China, and rapidly seeded multiple outbreaks across the globe. COVID-19 is a major pandemic with unprecedented public health burden and death toll worldwide. As of 19 April 2021, the SARS-CoV-2 pandemic has afflicted more than 141 million individuals worldwide, and has led to confirmed deaths of over 3 million people from 223 countries and territories [1]. 

Of the seven members of *Coronaviridae* family known to infect humans, SARS-CoV-2 is the third outbreak in less than two decades resulting in a major pandemic [2,3,4,5,6]. The other two major coronavirus acute respiratory disease outbreaks are SARS, caused by SARS-CoV (2002–2003), and Middle East respiratory syndrome (MERS), caused by MERS-CoV (emerged in 2012) [2]. SARS-CoV-2 is genetically closely related to SARS-CoV [2,6,7,8,9]; both exhibit an age-related increase in disease severity and mortality [10]. Whereas SARS-CoV-2 is associated with a significantly less crude case fatality rate (0.25–5%) [11], SARS-CoV and MERS-CoV were associated with high case fatality rates of ~10% and 34%, respectively [12]. The other endemic human coronaviruses—HKU1, NL63, OC43, and 229E—cause primarily common cold and contribute to 15% to 29% of common cold cases annually [13].

The speed of the scientific advances in understanding the biology of SARS-CoV-2 and the pathophysiology of COVID-19 as well as the development of effective vaccines will remain one of the greatest achievements of the human race. Within weeks, scientists were able to describe the clinical syndrome [3,4,5], identify SARS-CoV-2 as the causative agent [5,6], develop diagnostic tests [14,15,16], sequence the complete genome of the virus isolated from clinical samples [5], and start developing several vaccines candidates. Fast forward, in less than one year, we were able to successfully develop several efficacious vaccines; a few of them are now being used to vaccinate the general public [17,18]. In this review, we explore the available evidence on key virological, immunological, and clinical characteristics of SARS-CoV-2 infection with the emphasis on viral persistence, reinfection, and PASC. We highlight areas that warrant further investigation for the development of improved therapeutic and prevention interventions. 

## 2. SARS-CoV-2 and Its Tropism

Coronaviruses are large, enveloped, positive-sense, single-stranded RNA (+ssRNA) viruses that cause various diseases in mammals (Figure 1a). Human coronaviruses such as SARS-CoV-2 share a significant number of key features with other family members of the *Nidovirales* order, and primarily cause respiratory tract infections [2]. These key features of coronaviruses include (a) a highly conserved genomic organization that has a large replicase gene on the 5′ end encoding for several proteins with enzymatic activities, which is upstream of the structural and accessory genes (Figure 1b); (b) they all efficiently use ribosomal frameshifting to express their nonstructural proteins (nsps), polyproteins (Figure 1b), which are essential for in vivo replication; (c) as a member of the nidoviruses (*nidus* is Latin for “nest”), their downstream genes are expressed by synthesis of 3′ nested subgenomic mRNAs (Figure 1b). 

### 2.1. SARS-CoV-2 Origin

Human coronaviruses have an approximately 30 kb RNA genome and a diameter ranging from 50 to 150 nm with a distinctive crown-like appearance of their glycoprotein spike (S) protein (Figure 1a) [19,20]. These RNA viruses undergo genetic recombination, deletions, mutations, and other forms of variation allowing them to adapt to infect new hosts and spread within the same species independent of their natural reservoir. SARS-CoV-2 emerged in late 2019 and is ~80% and ~96% identical at the nucleotide level across its genome with SARS-CoV and the bat coronavirus, RaTG13, respectively [3,9]. Structural and biochemical characterization of the genome revealed that the receptor-binding domain (RBD) of the SARS-CoV-2 S protein is the most variable region with distinctive gain-of-function mutations to bind to the cognate cellular receptor, angiotensin-converting enzyme 2 (ACE2), with high affinity [3,5,9,21,22]. Furthermore, it was suggested that SARS-CoV-2 seemingly adapted through natural selection and genetic recombination in humans or other intermediate hosts such as the pangolin with high homology to the ACE2 receptor [3,6,9,23,24] before it emerged as a major pandemic in late 2019 in Wuhan, China.

SARS-CoV, MERS-CoV, and SARS-CoV-2 originated likely from bats and transmitted to human through intermediate hosts: SARS-CoV in 2002/2003 emerged through adaptation in palm Civets in a wildlife market in Guangdong, China [25], MERS-CoV in 2012 transmitted to human from dromedary camels in the Arabian Peninsula [26], but the definitive intermediate host for SARS-CoV-2 is unknown, and remains a source of controversy. Several groups reported that horseshoe bats as likely origin and imported pangolins as possible intermediate host for SARS-CoV-2. However, according to a comparative genomic analysis [27], none of these highly identical (90–96% identity at nucleotide level) coronaviruses that were found in these animals appear to be a direct progenitor of SARS-CoV-2, indicating that further adaptation in humans or other intermediate hosts must have taken place preceding the emergence of the COVID-19 pandemic [9].

### 2.2. SARS-CoV-2 Tropism

SARS-CoV-2 infects the target cell by binding to its cognate cell surface receptor, the ACE2 protein, using the RBD of the S protein (Figure 1c) [3,5,9,21,22,28,29,30,31,32]. The S proteins of SARS-CoV and SARS-CoV-2 are highly homologous and structurally similar with only minor differences, and they both use ACE2 to enter into target cells [3,21,28,30,31,33]. ACE2 is a type I transmembrane protein, which is ubiquitously expressed in endothelial and most epithelial cells of different organs, including in the high expressers such as the lungs, heart, kidneys, and small intestine [29,34,35,36,37,38,39]. It is regulator of the potent vasoconstrictor angiotensin maturation and hence offsets the vasomotor effect of ACE on the cardiovascular system [36]. 

Akin to all coronaviruses, the S protein of SARS-CoV-2 mediates two essential virus-host interactions in tandem during virus entry. First, it binds to the primordial and abundant sugar moiety with its N-terminus region, and second, it engages with high-affinity and selectivity to its protein receptor, ACE-2, to initiate fusion of the viral and host cell membranes via its C-terminal region (Figure 1c) [40]. These cascading events are believed to give coronaviruses an evolutionary advantage to easily adapt and expand their host ranges. SARS-CoV-2 virions entry into the cells is then orchestrated by proteolytic cleavage and activation of the ACE2-bound S protein by the type 2 transmembrane serine protease (TMPRSS2) on the cell surface [28,29,41,42,43]. After entry, through a series of highly regulated steps and virus–host interactions, the virus completes its replication cycle as described in Figure 1 [44]. Thus, a coordinated expression of ACE2 and TMPRSS2 is critical to augment viral entry and infectivity. Hou et al. [45], using a reverse genetics approach by single-cell RNA sequencing and in situ RNA mapping, showed an expression gradient of ACE2 in the normal airway epithelia, with the highest expression being in the nasal epithelial cells and lesser expression in the lower airway epithelial cells, terminating in significantly minimal ACE2 level in the alveolar pneumocytes. Interestingly, the expression of TMPRSS2 was not significantly different between the upper and lower airways’ epithelia [45,46]. However, the expression gradient of ACE2 and hence the very low frequency of dual ACE2^+^/TMPRSS2^+^ cells in the lower respiratory tract and alveoli likely translates to the early and greater SARS-CoV-2 replication in the upper airway than in the lower airway, and thus to the clinical course of the disease [47]. Germane to this scenario is that the SARS-CoV-2-induced high-IFN inflammatory microenvironment upregulates ACE2 expression without significantly affecting TMPRSS2 expression [45,46,48]. This may potentially increase the frequency of virus-susceptible dual ACE2^+^/TMPRSS2^+^ cells in the lower respiratory tract, although there is significant interhost variability in SARS-CoV-2 disease severity [45,46]. In fact, the degree of IFN-induced expression of ACE2 correlates with organ tropism, symptom onset, and severity of the SARS-CoV-2 illness [29,38,45,46,48].

During the early period of the pandemic, it was speculated that there is a notable upregulation of ACE2 by the potent antihypertensive drugs, ACE inhibitors may have contributed to the disproportionately increased susceptibility of patients with hypertension, heart diseases, and diabetes mellitus who are taking these drugs to SARS-CoV-2 infection and risk of developing a more severe disease. However, several studies have found no meaningful association between taking these drugs and risk of SARS-CoV-2 infection, disease severity, or mortality due to COVID-19-related illnesses [49,50,51,52,53]. Thus, the variability of SARS-CoV-2 susceptibility of the airway epithelial cells and disease course cannot fully be explained by ACE2 and TMPRSS2 expression levels, begging further in-depth analysis of the virus–host interactions and identification of major determinants. Indeed, one of the hallmarks of SARS-CoV-2 infection is that the virus hijacks a number of cellular factors and dampens the host immune response to facilitate its replication. After cell entry, SARS-CoV-2 compels the host cells to support its replication by co-opting different cellular factors and machineries. The molecular basis of acute SARS-CoV-2 infection, virus–host interactions, and immunopathogenesis has been extensively studied [4,5,21,28,29,30,31,32,34,37,38,43,45,46,47,48,54,55,56,57,58,59,60,61,62,63,64,65,66,67] and has led to the development of select therapeutics (see WHO guideline [68]) and successful vaccines [17,18].

## 3. COVID-19 Pathogenesis

SARS-CoV-2 is primarily a respiratory virus, and is transmitted by respiratory droplets, aerosols, and direct/indirect contacts. It can also infect other cell types and has been detected in a wide range of organs and tissues, notably the intestinal epithelium, liver, kidneys, brain, pancreas, eye, and immune cells [39,61] (Figure 2a). Such broader organ tropism is dictated by the expression of ACE2. The portal of entry is believed to be the nasopharyngeal or conjunctival epithelial cells that express high levels of ACE2 [29,45,57,69]. Although it is not clear whether the virus gains access to the terminal airways and alveoli by aspiration of the nasopharyngeal content, systemic dissemination, or by progressively moving distally on the bronchial tree mucosa, some viruses infect the alveolar epithelial cells and lung tissue-resident macrophages [55,69] (Figure 2b). Unlike SARS and MERS, the majority of individuals with SARS-CoV-2 remain asymptomatic or develop only mild symptoms [10,55,70,71] (Figure 3a). This is largely because of the clearance or control of the infection by the innate immune response effectors and virus-specific T cells without intensified tissue damage [60] (Figure 3b). The major pathophysiological feature and leading cause of death in COVID-19 patients is acute viral pneumonia characterized by bilateral ground glass opacities on imaging studies and histopathological features such as diffuse and organizing alveolar damage, proinflammatory infiltration, microvascular thrombosis, and persistence of SARS-CoV-2 in the respiratory tract [19,66,72,73]. 

A subgroup of patients with certain risk factors listed in Table 1 are associated with severity and increased fatality of COVID-19 [59,74,75,76]. The proxy for inflammation-induced multiorgan injuries is the presence of markedly elevated levels of proinflammatory biomarkers such as C-reactive protein (CRP), ferritin, IL-1, and IL-6 [74,77], and as such an unrestrained host immune response misfires to induce severe immunopathological changes, and hence multiorgan failure [19,38,46,55,62,66,69,72,78,79,80,81,82,83,84]. Su et al. [79] carried out an integrated clinical and multi-omics analysis on a large cohort of patients to understand the role of the hyperinflammatory state on the heterogeneity of severe COVID-19, and they described a distinct profile of proinflammatory cytokines associated with a loss of metabolites in severe COVID-19 compared to mild/moderate COVID-19. Such atypical immunophenotype signatures and stressed proinflammatory environments in severe SARS-CoV-2 infection were also observed by other research groups [65,78,79]. However, understanding the molecular crosstalk among direct viral cytopathicity, host immune dysregulation, and SARS-CoV-2-induced hypercoagulable states in SARS-CoV-2 viral sepsis and severe COVID-19 warrants further investigations.

### Molecular Pathology in Severe COVID-19

Acute respiratory distress syndrome (ARDS) in severe COVID-19 is characterized by loss of the epithelial–endothelial barrier integrity, capillary fluid leakage, diffuse alveolar damage, hyaline membrane deposition, and unrestricted recruitment of inflammatory effectors [19,66] (Figure 2b). Although the central role of lung pathologies as manifested by severe pneumonia and ARDS is the hallmark of severe COVID-19, extrapulmonary pathologies have also been consistently reported [19,96] (Figure 2a). The cytokine storm associated with severe SARS-CoV-2 infection has ripple effects across the body contributing to the viral sepsis, multiorgan failure, and even death [97]. SARS-CoV-2 hijacks the cellular machineries to facilitate its propagation and to induce a hyperinflammatory state [98]. The severity of SARS-CoV-2 infection is fueled by the dysregulation of the host immune response primarily by inhibiting type I interferon (IFN) response in acutely infected cells [65,78,83].

This hyperinflammatory state is marked by an accelerated viral replication associated with greater tissue damage by pyroptosis and coagulopathy, triggering a cascade of systemic immune response disequilibrium and a shift in the dynamics of host genes expression [63,78,82,84,96,97]. A recent analysis of the lung proteome showed a unique set of pathways that are required for SARS-CoV-2 replication in infected cells [96]. The analysis also identified important virus–host interactions that inhibit the function of host innate immunity-related genes. As discussed in Figure 1, the translation of nested subgenomic RNAs of SARS-CoV-2 requires capping, and the cap-recognizing host factor eukaryotic translation initiation factor 4E (EIF4E) is upregulated selectively in SARS-CoV-2-infected lung tissue. Furthermore, several genes involved in the antiviral innate immune response pathways, including the stress granule related factor Ras GTPase-activating protein-binding protein 1 (G3BP1), the mitochondrial protein translocase of the inner membrane 10 (TIM10), transcription regulators, ubiquitination pathway components, and the proinflammatory cytokine receptor interleukin 17 receptor A (IL17RA), were dysregulated in SARS-CoV-2-infected lungs [64,96].

The virus employs these host factors and pathways either directly or indirectly to promote its replication, which further facilitates the recruitment of proinflammatory cells and activation of the profibrotic pathways to incur severe tissue damage. For instance, Nie et al. [96] conducted an unbiased proteomic atlas using tissues obtained from different organs (lungs, kidneys, heart, testis, spleen, liver, and thyroid) at autopsy of COVID-19 cases. Of the 11,394 proteins they analyzed, 5336 involved in the host immune response, coagulation regulation, angiogenesis, and profibrotic processes were significantly perturbed during acute SARS-CoV-2 infection with an organ-specific niche, suggesting the existence of crosstalk among multiple organ systems during the exuberant immune response and accompanying tissue hypoxia, a phenomenon that was also described by others [78]. However, the heterogeneity of the host immune response, immunopathology, and clinical spectrum of SARS-CoV-2 infections in individuals with comparable viral loads remains a vexing question. For example, children and young adults with SARS-CoV-2 tend not to develop severe disease irrespective of viral load [71]. Alternatively, an intriguing imbalance between virus infectivity and the host immune response was demonstrated in a twin publication by Casanova and colleagues [76,99]; the authors uncovered over-representation of individuals with inborn errors in the type I IFN pathway in severe COVID-19 patients compared to either mild COVID-19 patients or healthy donors [76]. What is even more startling is that in the accompanying publication, the authors demonstrated that the presence of anti-type I IFN autoantibodies, presumably virus-induced, is linked with disease severity [99]. Although severe inflammation with aberrant immune activation is not unexpected in severe SARS-CoV-2 infection, these discoveries represent important inroads for future studies.

## 4. SARS-CoV-2 and Persistent Viral Shedding

### 4.1. Viral Dynamics and Duration of Infectiousness

Quantitative studies of SARS-CoV-2 viral load [47,59,100,101,102,103,104,105,106,107], dynamics [47,100,103,104,105,106,108], shedding [47,103,106,107], and viable virion isolation [47,103,109,110,111,112] have provided the following understandings into the pathogenesis of COVID-19: (i) the average incubation period (time from infection to symptom development) for SARS-CoV-2 is ~5 days (range 2–14 days), which is shorter than that for SARS and MERS [113,114,115,116]. (ii) Unlike SARS-CoV and MERS-CoV, SARS-CoV-2 viral load, and kinetics of shedding, is much higher in the upper than in the lower respiratory tract [101,108]. It is reported that the positivity rate was in fact higher for lower respiratory tract samples than it was for the nasopharyngeal swab [117]; however, the authors pointed out that their data could not be correlated with clinical symptoms or disease course. The demonstration of a strong concordance between nasopharyngeal swab and bronchoalveolar lavage (BAL) samples for SARS-CoV-2 RT-PCR test positivity established nasopharyngeal swab as the cost-effective and least invasive diagnostic method for SARS-CoV-2 [118,119]. (iii) The quick decline in the viral load in individuals with SARS-CoV-2 makes isolation and other prevention interventions challenging and less effective [116]. (iv) COVID-19 has a broad clinical spectrum (mild, moderate, severe, and critical COVID-19), and yet a great majority of individuals with SARS-CoV-2 are asymptomatic, and are as infectious as the symptomatic ones [116,120,121,122,123] (Figure 3a). A new report based on the analytical model and meta-analysis of select reports suggested that about 50% of new SARS-CoV-2 infections were contributed by exposure to either asymptomatic or presymptomatic individuals with SARS-CoV-2 [116]. This further emphasizes that successful control of the pandemic requires multipronged approaches including identification and isolation of asymptomatic and symptomatic cases, universal wearing of face masks, social distancing, contact tracing, and widespread use of therapeutics and/or vaccines. 

Several studies reported that on average the duration of viral particle shedding from the upper airway is ~17 days, the lower respiratory tract ~15 days, feces ~13 days, and blood ~17 days (reviewed in ref. [124]). The longest durations reported to date for detection of viral RNA were 83 days, 59 days, 126 days, and 60 days in the upper airway, the lower respiratory tract, feces, and blood, respectively [124,125]. Not surprisingly, SARS-CoV and MERS-CoV RNA were also detected from different samples such as upper respiratory tract, lower respiratory tract, serum, and stool weeks to months after recovery from the acute illness [126,127,128,129,130,131,132]. Despite prolonged detection of viral RNA, viable virus isolation was successful only in the first 8 days after symptom onset and the success of isolation of viable virions from respiratory samples directly correlates with the initial viral load [47,110,111,133] (Figure 3a). Because of the precipitous decline in viral load after the second week post-infection, transmission of SARS-CoV-2 was infrequent after the first 10 days of illness. Even though viral RNA was detected after a prolonged period in the blood samples, no replication-competent virus was isolated from PCR-positive blood samples [33], viable virus was isolated from fecal samples [134]. Nonetheless, fecal–oral route transmission of SARS-CoV-2 is yet to be demonstrated.

### 4.2. The Relevance of Prolonged SARS-CoV-2 Viral Shedding

#### 4.2.1. Role of SARS-CoV-2 Organotropism and Immune Privilege in Viral Persistence

Viruses are highly sophisticated molecular machines that can undergo intra-host evolution to develop effective strategies to overcome host immunity and establish chronic infection by replicating continuously, establishing latent reservoir, or integrating into the host cell genome. If the immune system cannot eliminate the virus, the chronic immune activation and/or the cytopathic effect of the virus will continue until the infection resolves or kills the host. Certain viruses chronically persist by establishing metastable virus–host immune response interaction equilibrium. For example, viruses such as the human immunodeficiency virus (HIV), hepatitis B virus (HBV), and hepatitis C virus (HCV) compel the host cell to persist in latent state and evade the host antiviral immunity [135]. Other viruses like Ebola virus taught us that persistence in immune-privileged sites can also lead to transmission to a new host via an unexpected and different route [136]. Whether SARS-CoV-2 can establish chronic infection or persist in immune-privileged anatomic sanctuary sites remains to be demonstrated.

SARS-CoV-2 infection is associated with accelerated replication and high viral load during the acute phase, which is followed by a rapid decline after the first week [47]. In some instances, the peak of the viral titer in the respiratory tract occurs during the presymptomatic period [137]. Interestingly, analysis of autopsy samples from severe COVID-19 patients revealed that viral RNA can be detected until the time of death, suggesting that prolonged shedding of virus may indicate a grave outcome [85]. SARS-CoV-2 nucleic acids persistence during convalescence and the risk of infectiousness remain controversial. The lingering detection of SARS-CoV-2 RNA in different clinical specimens of convalescent patients could be due to slow replication, reactivation of latent virus, or residual nonreplicating genetic material; however, it needs to be systematically explored to definitively determine its relevance. Uncovering the transition of the disequilibrium among the exuberant antiviral immune response, rapid SARS-CoV-2 replication during the acute phase, and severe immunopathology to reset the virus–host interaction and establish persistent infection, if any, will be an interesting avenue of basic science and clinical research (Figure 4).

Viruses infect specific cell types and persist with little to no viral gene expression by sabotaging the cellular gene expression machineries to subvert sterilizing immunity. They can also establish persistent infection in immune-privileged tissues and evade constant immune surveillance. The respiratory tract epithelial cells and alveolar macrophages are the primary targets for SARS-CoV-2 replication. However, the cellular and organ tropism of SARS-CoV-2 is expanding [39,61], and whether some of these SARS-CoV-2-targeted immune-privileged sites can serve as sites of viral persistence remains an area of future investigation.

#### 4.2.2. Mechanism of SARS-CoV-2 Evasion and Subversion of Host Immunity

SARS-CoV-2 has a limited array of genes with approximately 14 open reading frames to encode 29 different proteins [138] (Figure 1b). Gordon et al. [138] mapped the SARS-CoV-2 interactome and uncovered the role of most of the viral proteins, which manipulate the host cell biological processes and factors. Some exhibit specialized and niche-specific roles to evade or subvert the immune response. For instance, Nsp9, Nsp13, Nsp15, Orf3a, Orf9b, and Orf9c were identified to interfere with the IFN pathway and proinflammatory cytokine expressions; others disrupt the adaptive immune response, gene expression and protein translation machineries [138]. Another large-scale multilevel study showed that SARS-CoV-2 disrupts different cellular functions, including transcriptome, proteome, ubiquitinome, and phosphoproteome [139]. Identifying and targeting the vulnerability hotspots of SARS-CoV-2 and mechanisms of immune evasion are promising research areas for further research and development of antiviral agents. 

#### 4.2.3. Viral Persistence in Immunocompetent Hosts

Although understanding the kinetics and dynamics of viral shedding in relation to infectivity is critical in implementing infection prevention and control strategies, it has been a futile endeavor to demonstrate a correlation between prolonged detection of viral RNA and isolation of viable virus from clinical samples [47,111,140]. A recent meta-analysis identified a correlation between prolonged SARS-CoV-2 shedding as determined by RT-PCR and the increased risk of developing severe disease [141]. When the authors stratified the pooled median duration of respiratory RNA shedding by disease severity, they found that the median duration of nasopharyngeal viral RNA shedding was 19.8 days among severely ill patients and was 17.2 days in mild-to-moderate COVID-19 cases [141]. To define the SARS-CoV-2 transmission window, Sai et al. [142] carried out a COVID-19 pathogenesis and transmissibility study using a hamster SARS-CoV-2 infection model and found a strong correlation between transmission and isolation of viable virus in cell cultures. Interestingly, they found no correlation between prolonged detection of viral RNA from the nasopharyngeal swab and transmissibility, further corroborating that infectivity declines precipitously and prolonged detection of viral RNA does not necessarily mean that the individual is infectious. The authors also demonstrated the existence of an inverse correlation between infectiousness and development of neutralizing antibodies, an observation corroborated by others in recovered COVID-19 patients [143] as well as in a nonhuman primate model [144,145]. The median duration of infectiousness of the wild-type strain was determined to be approximately 8 days post onset of symptoms [47,111,143], and using probit analysis, van Kampen et al. [143] showed that the probability of detecting viable virus from respiratory specimens is ≤5% when the duration of symptoms is more than two weeks. To this end, there is only one report of successfully isolated infectious virus from the nasopharyngeal swab 18 days after symptoms onset [140], confirming that infectiousness drops dramatically after the second week. It appears that detection of subgenomic viral RNA is a poor predictor of infectiousness and often outlasts the detection of replication-competent virus, and the prolonged detection of viral RNA could be due to the suboptimal immunological clearance of SARS-CoV-2 infection [47,111,146].

High SARS-CoV-2 viral load, duration of symptoms less than 7 days, absence of neutralizing antibodies, and host immunocompromization were proposed as potential predictors of prolonged shedding of infectious virus from respiratory tract. However, only high viral load >7 Log_10_ RNA copies/mL in respiratory samples and absence of SARS-CoV-2-neutralizing antibodies were found to be independent predictors of infectiousness [47,111,143]. These studies were based on data gathered from hospitalized severe COVID-19 patients. Knowing that the great majority of individuals with SARS-CoV-2 are either asymptomatic or exhibit only mild disease symptoms, a proper and evidence-driven public health risk stratification and intervention cannot be employed using these reports only. To address this issue, in a recent retrospective analysis using data of nonhospitalized patients from 282 transmission clusters, Marks et al. [147] confirmed that high viral load is indeed a major predictor of infectiousness of index cases. The author further suggested that, irrespective of the clinical spectrum of COVID-19, determination of viral load and reinforcement of contact tracing and quarantine policy will remain crucial in devising and revising the pandemic control policies. It is also important to understand why some, but not all, COVID-19 patients continue to shed viral particles for a prolonged period, as cost-effective stratification and intervention strategies are the cornerstone of our public health effort in controlling the pandemic. 

#### 4.2.4. Viral Persistence in Immunocompromised Hosts

A recent analysis of the literature on the outcomes of SARS-CoV-2 infection in individuals with a wide range of conditions found that lung and hematologic malignancies, solid-organ transplantation, and primary immunodeficiencies put the patient at higher risk for (i) nosocomial acquisition of SARS-CoV-2, (ii) development of severe COVID-19, and (iii) COVID-19-related death compared with immunocompetent individuals [148]. On the other hand, those on biologics and immunosuppressive therapy appear to not be at increased risk of developing severe COVID-19 [92,148,149], an evidence further supporting the role of unrestrained immunity in disease severity and the potential role of inflammation modulators in treating COVID-19, including steroids [150]. In sub-Saharan Africa where the burden of another viral illness, HIV/AIDS, is paramount, the concern was if HIV-SARS-CoV-2 co-infection results in greater severity and higher mortality of COVID-19. In fact, a few large-scale retrospective analyses indicated that people living with HIV are at higher risk of developing and dying from severe COVID-19 [151,152,153]. Whether this holds true, it is important to carry out studies by comparing controlled versus uncontrolled HIV infection and by analyzing the role of other comorbidities, which are likely to bias the outcome of any meaningful interactions between the two pathogens in co-infected individuals [154,155].

Bridging these knowledge gaps will allow us to understand the pathophysiology of SARS-CoV-2 infection in immunocompromised individuals and develop patient-tailored interventions. Furthermore, this allows us to explore the interplay between preexisting immunosuppression and SARS-CoV-2 infection, with an emphasis on COVID-19 clinical course and outcome, identification of atypical manifestations, and the effect of immunosuppressants on SARS-CoV-2 replication in this special group of at-risk population. The few case reports suggest that there is a high likelihood of viral persistence, reactivation, or re-infection in immunocompromised patients, which also warrants further exploring.

As discussed above, prolonged shedding and recurrent detection of viral RNA by RT-PCR from the clinical samples of asymptomatic patients with SARS-CoV-2 as well as from symptomatic patients have been well documented. However, there is very little success in isolating infectious virus after 1–2 weeks of symptom onset in immunocompetent patients [124]. Interestingly, there are several case reports demonstrating isolation of infectious viral particles from immunocompromised patients for weeks to months. Some of them presented with relapse of the symptoms [156,157,158,159,160,161]. Others required multiple courses of treatment with remdesivir [162], and ultimately succeeded to clear the virus [156,159,161]. The longest duration reported so far to isolate viable virus from nasopharyngeal swab was ~8 months, which was from a non-Hodgkin’s lymphoma patient [161]. One can speculate that the selective pressure in an immunocompromised patient is likely to be different from that in immunocompetent individuals. However, it is difficult to extrapolate and generalize based on the few case reports discussed here on SARS-CoV-2 persistence and prolonged shedding of infectious virus, for the following reasons: (i) the phylogenetic analyses of these limited case reports were not carefully designed to explore factors that influence the risk of SARS-CoV-2 persistence or reinfection; (ii) these case reports involve an atypical disease course of SARS-CoV-2 infections since the majority of infections are classically acute in nature, arguing against the establishment of sustained infections and thus limiting generalization based on the reported intra-host viral genetic diversity in these cases; (iii) the rate of SARS-CoV-2 evolution is slow [163], an extra Achilles heel to the diversification of SARS-CoV-2 viral quasispecies. However, because the SARS-CoV-2 pandemic is a widespread infection, it is possible that the host immune pressure could shape SARS-CoV-2 evolution by contributing to genetic diversity and selection, and thus contributing to phenotypic changes and establishment of chronic infections. This dimension of SARS-CoV-2 infection warrants large-scale studies designed to understand the underpinning mechanism by defining temporospatial resolution of SARS-CoV-2 persistence to help develop potential patient-tailored therapeutics and preventive strategies. In light of the accumulating and emerging evidence on the effect of the new SARS-CoV-2 variants on efficacy of vaccines, their transmissibility, and severity of the disease [164], it would be also interesting to investigate the relationship between intra-host viral evolution and the mechanism of immune evasion in different patient groups in the context of viral persistence, infectiousness, and reinfection.

## 5. SARS-CoV-2 Reinfection

SARS-CoV-2 reinfection is rare, but there are a few case reports of reinfection that were confirmed by whole-genome sequencing and phylogenetic analysis [165,166,167,168,169,170,171,172,173]. It is also important to note that these reports are likely to represent a small fraction of reinfections. The corollary of pathogen invasion is that the host attempts to control the primary infection and generates protective immune memory to combat future re-emergence/reinfection, the basis of vaccination strategies. Likewise, we expect that the majority of individuals who recovered from primary SARS-CoV-2 infection developed protective B cell and T cell immunity, even if the longevity of protective immune memory is unclear [174]. Indeed, Lumley et al. [174] analyzed longitudinal cohort data to determine the incidence of SARS-CoV-2 reinfection in healthcare workers, and they were able to uncover the relationship between seropositivity during primary infection and incidence of reinfection over a six-month period of follow up during convalescence. The authors found a strong inverse association between the presence of high anti-spike IgG titer and reinfection, reinforcing the observations in earlier reports by other researchers [144,145,175,176]. Irrespective of the detection of subgenomic viral RNA in the nasopharyngeal swab in select patients and animal models, the presence of an anamnestic immune response after primary infection provides potent protection against symptomatic SARS-CoV-2 reinfection [177]; a critical insight into the durability of immunotherapies and vaccine protection as well as the timing of booster doses.

The heterogeneity of the COVID-19 clinical spectrum and host antiviral immune responses coupled with the intrinsic viral evasion mechanism(s) highlight the need to identify factors that predict not only disease severity but also factors that drive the generation of new viral variants. SARS-CoV-2 continues to spread, and the effects of evolutionary pressure are revealing viral adaptation and escape variants, which are highly infectious and potentially more fatal. Some of the emerging mutants are summarized here [178]. As new variants continue to emerge with mutations in the S protein, the growing concern will be their gain-of-function to evade protective immunity afforded by the currently available vaccines and leading to a widespread reinfection. Harrington et al. [179] reported the first case of reinfection by one of the most virulent variants, VOC-202012/01, after approximately 8 months post-primary infection, causing severe COVID-19 during his second infection. Moreover, the recent phase 1b/2 trial that was conducted in South Africa assessed the efficacy of the AstraZeneca ChAdOx1 nCoV-19 vaccine (AZD1222) against the original D614G and the B.1.351 variants (see Table 2), and the vaccine did not protect against the B.1.351 variant infection, suggesting that reinfection by emerging variants is conceivable in immune individuals [180]. The number of new SARS-CoV-2 variants is expanding, and the variants listed in Table 2 are of significant public health consequences in regard to their potential impact on reinfection rate, the efficacy of currently available vaccines, and the design of newer versions of vaccine, if needed, in the future.

As these and other reports of reinfection surface, understanding and integrating the correlates of protective immunity and intra-host evolution as determined by genomic surveillance need further studies to document and help control the pandemic both at local and global levels. Robust data on the duration of protective immunity and viral shedding, the implication of reinfection and infectiousness of the new variants, the role of antibody-dependent enhancement with the new variants in the setting of prior vaccination, and the efficacy of the currently available vaccines against the emerging SARS-CoV-2 variants are all crucial for informed decision-making and policy revision. Because of the dynamic nature of the SARS-CoV-2 spread, the expansion of the ongoing population-based genomic surveillance initiatives in various countries will be beneficial since reinfection and reintroduction will accelerate the molecular clock of the emergence of more virulent and/or resistant variants worldwide, which could potentially affect the efficacy of the current effective vaccines developed using the wild-type strain.

## 6. The Post-Acute Sequalae of SARS-CoV-2 Infection (PASC)

The SARS-CoV-2/COVID-19 pandemic has rapidly engulfed the world with nearly 120 million confirmed infections and over 2.6 million associated fatalities as of 15 March 2021 [1]. Compelling evidence is amassing and we are now beginning to understand the aftermath of SARS-CoV-2 infection referred to as “Long COVID”, which is experienced by ~10% of SARS-CoV-2 infection survivors [190,191]. Although there is no consensus on the definition of “Long COVID”, most of the reports defined the syndrome as a cluster of symptoms that persisted for more than 4 weeks following a confirmed or suspected SARS-CoV-2 infection [191]; others used 2 months as a clinical cut off to define “Long COVID” [192,193].

Recently, the NIH renamed “Long COVID’ as post-acute sequalae of SARS-CoV-2 infection (PASC) and launched a comprehensive research initiative to understand the sequalae of SARS-CoV-2 infection [194] (Figure 3). PASC is a syndrome in COVID-19 survivors that present with a plethora of symptoms, including cough, dyspnea, fatigue, myalgia, dysautonomia manifested as postural orthostatic tachycardia syndrome, gastrointestinal symptoms, dermatological, or neurological symptoms [191,192,193,195]; symptoms should either persist beyond 4 weeks after their onset or newly develop as sequelae of COVID-19. To this end, the NIH multipronged initiative on PASC is the first of its kind, which will hopefully provide critical data on the natural history of PASC at a population level and will shed light on the potential long-term interventions to be employed in managing COVID-19 survivors.

Long-term complications are not unexpected given the broader organotropism of SARS-CoV-2 and severity of the tissue injuries [61]. However, the pathophysiology and natural course of PASC is unclear, meriting further studies. Some potential mechanisms may include (i) the cytokine storm, (ii) the immune misfiring and virus-triggered autoimmunity, (iii) the overwhelming systemic endotheliitis resulting in chronic microangiopathy and thromboembolic events, (iv) persistent viral RNA shedding triggering chronic immune activation, and (v) finally, the possible role of viral persistence, reinfection, and reintroduction of new variants [192,196]. All of these may contribute to PASC pathogenesis either individually or in combination. Therefore, it is vital to recognize the magnitude, complexities, and heterogeneities of COVID-19 illnesses and to continue to support better research to develop long-term multidisciplinary care and rehabilitation for COVID-19 patients during convalescence.

## 7. Conclusions

The unprecedented nature of the COVID-19 pandemic has demanded an urgent and unmatched productivity by all stakeholders, and the scientific community has met the call with a historic proficiency to develop effective vaccines in less than one year. SARS-CoV-2 is the third outbreak in the last two decades, and it exhibits some differences compared to its predecessors SARS-CoV and MERS-CoV. This review summarizes the rapidly evolving important discoveries on SARS-CoV-2 with an emphasis on the mechanism and relevance of viral persistence and PASC. We also highlight potential areas of research on viral persistence and PASC. Addressing these issues will provide a robust framework to fulfill the unmet needs in our efforts to control this pandemic and enable readiness for any future pandemics.

## Figures and Tables

**Figure 1 viruses-13-01025-f001:**
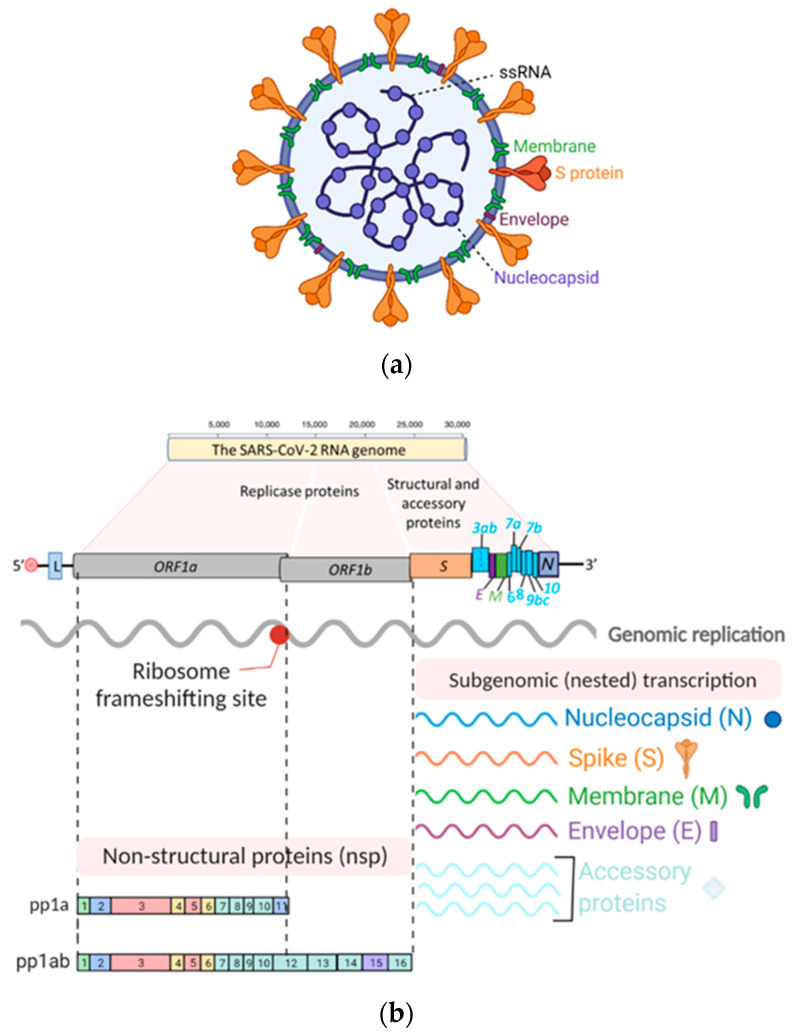
SARS-CoV-2 genome organization and replication cycle. (**a**) The SARS-CoV-2 virion structure. While the viral membrane and envelope proteins ensure its genomic RNA incorporation and assembly, the trimeric spike (S) protein provides specificity and high-affinity binding for its receptor to enter into target cells. The positive-sense, single-stranded RNA genome (+ssRNA) is encapsidated by the nucleocapsid. (**b**) Schematic depiction of SARS-CoV-2 genome architecture and the poly-(proteins) it encodes. (**c**) Schematics of SARS-CoV-2 replication: (1) SARS-CoV-2 virions bind to ACE2, its cellular receptor, and the type 2 transmembrane serine protease (TMPRSS2), a host factor that promotes viral particles entry and fusion at the plasma membrane or endosomes. (2) In the cytosol, the incoming gRNA will be released and subjected to immediate translation of the ORF1a and ORF1b open reading frame resulting in the polyproteins (pp1a and pp1ab), which are further proteolytically processed to the components of the replicase complex. (3) The replicase complex is an assemblage of multiple nonstructural proteins (nsps) that orchestrate transcription of the viral RNAs and viral replication. Concordantly, the virus compels the host cell to create the viral replication organelles, including the perinuclear double-membrane vesicles (DMV), the convoluted membranes, and small, open double-membrane spherules (DMS). These structures provide a protective microenvironment for the viral gRNA and subgenomic mRNA during replication and transcription. (4,5) The newly synthesized surface structural proteins translocate to the ER membrane and transit through the ER-to-Golgi intermediate compartment (ERGIC), which then assembles and encapsidates the ribonucleocapsid. (6,7) Ultimately, the progeny virions bud into the lumen of the secretory vesicles and will be released by exocytosis.

**Figure 2 viruses-13-01025-f002:**
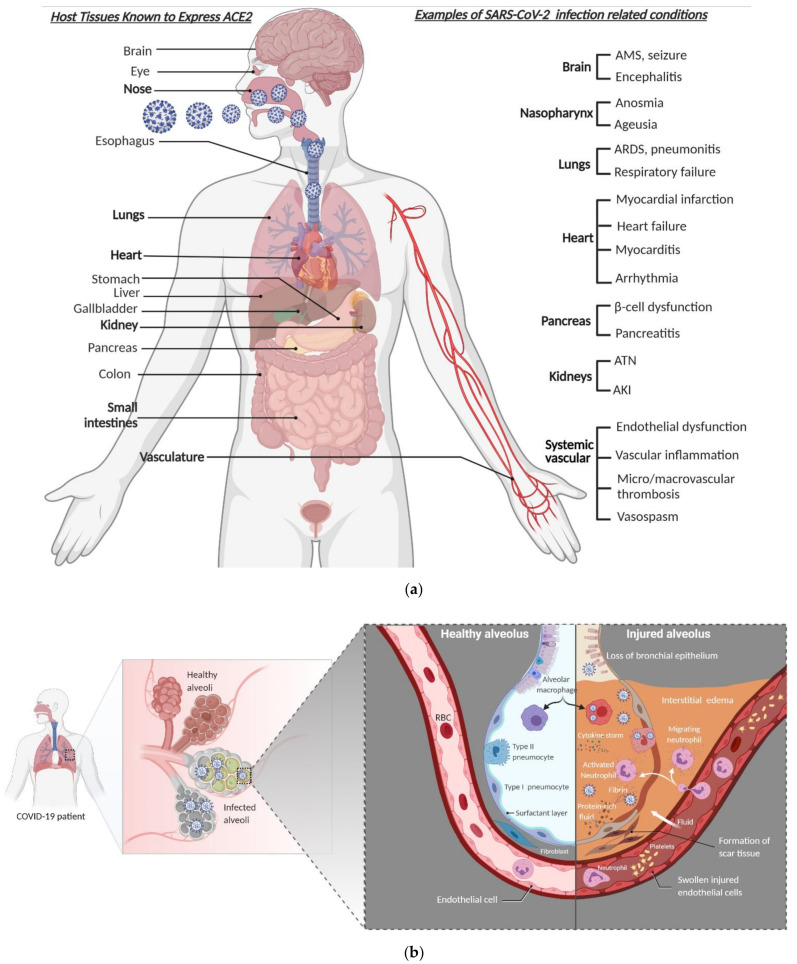
SARS-CoV-2 tropism, clinical presentation, and pathophysiology of alveolar damage. (**a**) A wide range of clinical and laboratory abnormalities can be observed in SARS-CoV-2 infection. In severe COVID-19 cases, the pathophysiology of the disease involves accelerated viral replication, cytokine storm, hyperinflammatory state, systemic endotheliitis, and hypercoagulability with secondary organ dysfunction (often pulmonary, cardiovascular, renal, or hepatic). (**b**) When SARS-CoV-2 makes it to the lower airway and infects type II pneumocytes (right side), the accelerated replication of the virus induces host cell death by pyroptosis and release of proinflammatory and damage-associated molecules, further activating and recruiting proinflammatory immune cells, including neutrophils, lymphocytes, and monocytes to the site of infection. In a defective or misfiring immune response, there will be an overproduction of proinflammatory and profibrotic cytokines, resulting in diffuse alveolar damage with fibrin-rich hyaline membrane deposition and loss of the gas exchange function of the alveoli, and hence development of hypoxic respiratory failure. Furthermore, the resultant cytokine storm leads to multi-organ damage. However, in a healthy immune response, virus-specific immune cells will be recruited and eliminate the infected cells before the virus spreads to ultimately minimize the alveolar damage (the left side). AMS denotes altered mental status, ARDS denotes acute respiratory distress syndrome, AKI denotes acute kidney injury, and ATN denotes acute tubular necrosis.

**Figure 3 viruses-13-01025-f003:**
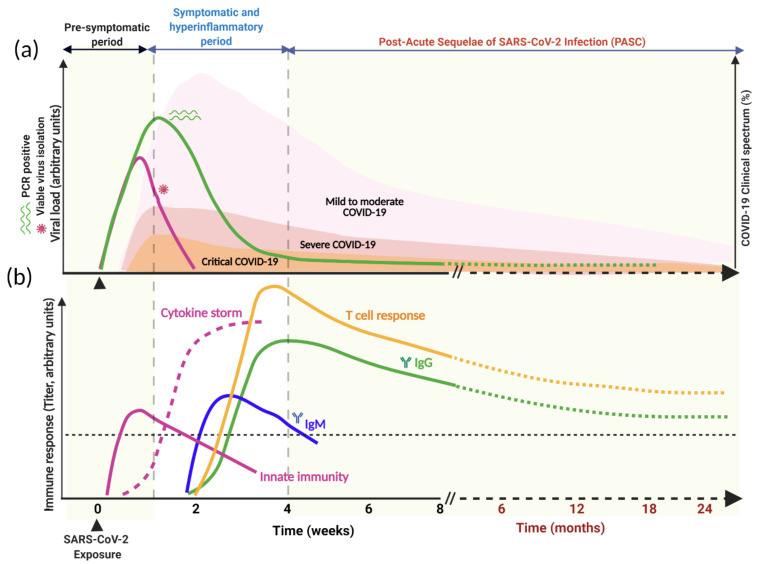
Time course of SARS-CoV-2 infection, clinical spectrum, and immune response. (**a**) Kinetics of viral load and clinical spectrum. (**b**) A schematic representation of immune response following SARS-CoV-2 exposure. The innate immune response appears shortly after exposure (purple line) accompanied by cytokine storm (purple dashed line), followed by adaptive immunity development after the first week post-infection. Seroconversion and T cell response occur during the second week after symptoms onset. While IgM peaks early and decays faster, IgG titers and T cell response detected around day 10 and their peak and height are likely to be influenced by disease severity and virus load. The level of antibody protection from reinfection, the duration of the total humoral immune response above this protective threshold (dashed black horizontal line), and the rate of decline from mild- or severe-infection-induced antibodies is not known for SARS-CoV-2 during the PASC period.

**Figure 4 viruses-13-01025-f004:**
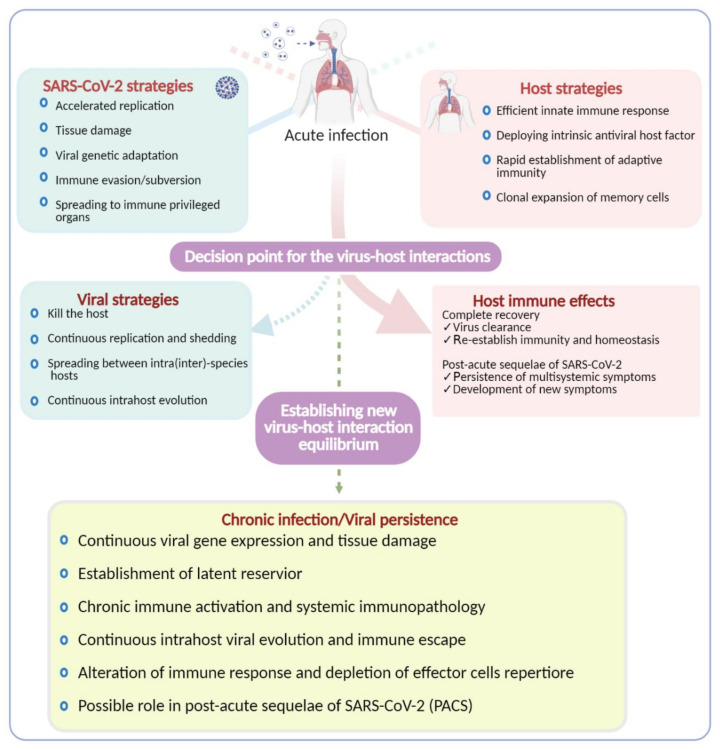
Outcomes of SARS-CoV-2 infection. During the acute infection, the host and the virus compete for dominance to tip the balance in their favor, resulting in different outcomes as indicated in the highlighted boxes. If the host strategies prevail to achieve complete recovery, the virus will be cleared. Another outcome that is poorly understood and yet getting traction is the multisystemic sequelae of SARS-CoV-2 infection called PASC. On the other hand, if the virus wins the battle, a variety of outcomes are expected, as indicated in the illustration. Viral persistence in a minority of patients, particularly in immunocompromised patients, has now been well documented; however, the underlying mechanism is unknown. There is a possibility of establishing metastable equilibrium between the host and SARS-CoV-2 to facilitate persistence and/or chronic infection.

**Table 1 viruses-13-01025-t001:** Risk factors for severe COVID-19 and worse outcomes.

Factors ^1^	References
Male sex	[59,74,75]
Advanced age	[59,74,75,85]
Chronic respiratory diseases (asthma, COPD, ILD)	[75,86]
Major cardiovascular disease (CAD, CHF, CMP)	[74,75,85]
Hypertension	[74,75,85]
Diabetes mellitus	[74,75,85,87]
Obesity (body mass index ≥ 30)	[75,88]
Chronic liver disease	[75,89]
Chronic kidney disease	[75,90]
Cancers	[75,91,92]
Immunocompromised state from solid-organ transplantation	[75,93]
Immunosuppressive therapies, Rituximab, and JAK inhibitors	[94,95]

^1^ Abbreviations: COPD, chronic obstructive pulmonary disease; ILD, interstitial lung diseases; CAD, coronary artery disease; CHF, congestive heart failure; CMP, cardiomyopathy; JAK, Janus kinase.

**Table 2 viruses-13-01025-t002:** Emergent SARS-CoV-2 variants of high consequence.

Name of Variants (Pangolin Lineage ^1^)	Key S Protein Substitutions	Origin	References
Current dominant variant	D614G ^2^	Europe	[181]
B.1.1.7	Δ69/70, Δ144Y, N501Y A570D, P681H	UK	[182]
B.1.351	K417N, E484K, N501Y	South Africa	[183,184]
B.1.427/B.1.429	S13I, W152C, L452R	USA (California)	[185]
B.1.526	T95I, D253G, E484K	USA (New York)	[186]
P.1	K417T, E484K, N501Y	Japan, Brazil	[187,188]

^1^ SARS-CoV-2 Pangolin lineages nomenclature was proposed by Rambaut et al. [189]; ^2^ One of the first mutations identified, which since April 2020 has become the dominant variant. All the other variants listed in the Table share the D614G mutation, which is believed to help them spread more quickly than viruses without this mutation [181].

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
