# Peer review of "Insights into SARS-CoV-2 Persistence and Its Relevance"

_viruses, 2021, doi:10.3390/v13061025_

Round 1

Reviewer 1 Report

Due to the enormous volume of SARS-CoV-2 literature, comprehensive and insightful reviews are desired by many researchers. The authors put a lot of efforts in this review article. The figures and tables are particularly helpful. Here are several comments for consideration.

Line 17 the statement of a living literature review is better to be made specific, by providing the period of time, when the most references were reviewed. Clearly only a limited number of papers published in 2021 were included.

Line 24 emerged as zoonotic spillover in Wuhan can be simply revised as first emerged in Wuhan. The origin of the virus is yet to be discovered.

Line 262 the section 4 is the most important part of the article. Though many references were cited, the critical reviewing of the reports is lacking. Considering the very large number of cases, to have prolonged infections and re-infections are not surprising. Whether these cases represent true and unique characteristic of COVID-19 and what are the trends across the extended pandemic. Putting more perspectives into the review will certainly be helpful and intriguing to readers.

Author Response

Response to Reviewer 1 Comments

Due to the enormous volume of SARS-CoV-2 literature, comprehensive and insightful reviews are desired by many researchers. The authors put a lot of efforts in this review article. The figures and tables are particularly helpful. Here are several comments for consideration.

Point 1: Line 17 the statement of a living literature review is better to be made specific, by providing the period of time, when the most references were reviewed. Clearly only a limited number of papers published in 2021 were included.

We would like to thank the reviewer for pointing out this point. Yes, we have not included the most recent and newest materials from 2021 as we planned to review the reports in the first year of the pandemic from Jan 2020 to early March 2021. However, we have included a few selected reports, particularly in the latter sections, including the references cited in the tables through early April, 2021. As suggested by the reviewer, we have now included the time period in page 1, line 18.

Point 2: Line 24 emerged as zoonotic spillover in Wuhan can be simply revised as first emerged in Wuhan. The origin of the virus is yet to be discovered.

We agree, and we have made the suggested modification in the manuscript on page 1, line 26.

Point 3: Line 262 the section 4 is the most important part of the article. Though many references were cited, the critical reviewing of the reports is lacking. Considering the very large number of cases, to have prolonged infections and re-infections are not surprising. Whether these cases represent true and unique characteristic of COVID-19 and what are the trends across the extended pandemic. Putting more perspectives into the review will certainly be helpful and intriguing to readers.

As suggested by the reviewer, we have included a number of reports in Section 4, and provided a balanced discussion and forwarded perspectives on this controversial and yet to be systematically studied issue. Because of the nature and rapid evolution of the SARS-CoV-2 literature and the fact that most of the reports included are case reports, we felt it would be unjust to provide a unifying perspective on persistence and reinfection. Therefore, we chose to provide a balanced perspective and area of future research areas under each subsection. Nevertheless, as pointed out by the reviewer, we have now expanded the summary paragraph on page 11, by adding extra points  in lines 484-503.

Reviewer 2 Report

The submitted review manuscript 'Insights into SARS-CoV-2 Persistence and Its Relevance' gives an excellent, surprisingly comprehensive, up-to-date overview of a large, complex and fast-moving field. It impressively covers not only basic virology but disease manifestations, pathophysiology, infection dynamics, impact of co-morbidities and re-infection with excellent discussion of controversial aspects of these areas in a thoughtful and thorough manner. At each point, it also concisely lays out all the key questions that remain to answered about the virus and its infection, the host response and, to an extent, potential future trajectory of the SARS-CoV2 pandemic.

This review is excellently written and a superb resource to anyone trying to keep up to date with this often confusing field. It's also one of the rare example of a manuscript I have reviewed with no typos I could identify. The diagrams/tables are also excellent, very thorough and high quality, perfectly assisting the text in summarising the field very well.  

Minor points:

Line 81 - 'originated from bats' - include 'likely' : although comparative sequence evidence suggests this, we don't have definitive evidence that SARS-CoV2 emerged from bats

160 - just ACE2? You noted just before this that a gradient of co-expression of both ACE2 and TMPRSS2 dorrelates with infectivity in the URT. Is there any evidence of TMPRSS2 expression also correlates here too?

Author Response

Response to Reviewer 2 Comments

The submitted review manuscript 'Insights into SARS-CoV-2 Persistence and Its Relevance' gives an excellent, surprisingly comprehensive, up-to-date overview of a large, complex and fast-moving field. It impressively covers not only basic virology but disease manifestations, pathophysiology, infection dynamics, impact of co-morbidities and re-infection with excellent discussion of controversial aspects of these areas in a thoughtful and thorough manner. At each point, it also concisely lays out all the key questions that remain to answered about the virus and its infection, the host response and, to an extent, potential future trajectory of the SARS-CoV2 pandemic.

This review is excellently written and a superb resource to anyone trying to keep up to date with this often-confusing field. It's also one of the rare examples of a manuscript I have reviewed with no typos I could identify. The diagrams/tables are also excellent, very thorough and high quality, perfectly assisting the text in summarizing the field very well.  

Minor points:

Point 1: Line 81 - 'originated from bats' - include 'likely' : although comparative sequence evidence suggests this, we don't have definitive evidence that SARS-CoV2 emerged from bats

We agree and have made the suggested modification on page 2, line 83.

Point 2: 160 - just ACE2? You noted just before this that a gradient of co-expression of both ACE2 and TMPRSS2 correlates with infectivity in the URT. Is there any evidence of TMPRSS2 expression also correlates here too?

We thank the reviewer for pointing out this critical point. We have made the necessary modification and clarification in the manuscript by rewriting and reorganizing the section on page 5, lines 134-175.